# Exploiting Parallelism for Fast Feynman Diagrammatics

**John Sturt$^{1\star}$ and Evgeny Kozik$^{1\dagger}$**

**1** Department of Physics, King's College London, Strand, London WC2R 2LS, United Kingdom

$\star$ john.1.sturt@kcl.ac.uk ,   $\dagger$ evgeny.kozik@kcl.ac.uk

## Abstract

**We achieve a dramatic acceleration of evaluating Feynman's diagrammatic series by use of specialised hardware architecture within the recently introduced combinatorial summation (CoS) framework. We present how exploiting the massive parallelism and concurrency available from GPUs leads to orders of magnitude improvement in computation time even on consumer-grade hardware, in some cases doubling the range of attainable diagram orders. This provides a platform for making probes of novel phenomena of strong correlations much more accessible.**

# 1 Introduction

Numerical techniques are imperative tools for quantum many-body systems used for calculating correlation functions, thermodynamic quantities, and scattering cross-sections in many fields including condensed matter, statistical, and high-energy physics. Despite the strength of contemporary techniques and high performance computing (HPC) infrastructure, calculating these quantities often remains a prohibitively costly and time-intensive task. In the context of condensed matter systems, state-of-the-art unbiased methods, such as Quantum Monte Carlo (QMC), tend to suffer due to finite size effects and the fact that the type of fermion physics usually of interest potentiates the sign problem [1, 2]. Diagrammatic Monte Carlo (DiagMC) methods [3–6] address some of these issues by representing the quantities of interest as sums of connected Feynman diagrams [7] constructed directly in the thermodynamic limit and evaluated exactly by stochastic sampling. The only systematic error in DiagMC is thus due to the truncation of the diagrammatic expansion at some large order $n_*$. However, the number of terms in the expansion increases factorially with $n_*$, and although there are algorithms for efficient summation of the integrands over all diagram topologies [8–10], the lowest computational cost of evaluating the series to a given relative statistical error is still exponential in $n_*$.

This is not catastrophic by itself because, for a convergent series, the computational cost is compensated by the exponentially decreasing with $n_*$ value of the residual contribution, which formally translates into a *polynomial* scaling of the computational time with the inverse of the required error bound [11] (and likewise for a divergent but resummable series [12]). In practice, however, the accessible $n_*$ must be large enough for the relevant physics to be captured and the asymptotic polynomial scaling to set in before the exponential wall makes the calculation impossible. Much progress has been achieved recently by algorithmic improvements that pushed the truncation order $n_*$ beyond the threshold for a reliable solution of several longstanding many-electron problems, ranging from pseudo-gap physics [13–16, 16, 17] to quantum magnetism [18, 19], the $SU(N)$ Hubbard model [10] and homogeneous electron gas [20, 21].

Here we demonstrate how adapting state-of-the-art Feynman diagrammatics for a dedicated hardware architecture—graphics processing units (GPUs)—can unleash several orders of magnitude speed-ups by itself, doubling the typically accessible diagram orders, and opening the door for reaching new physics and developing novel diagrammatic methods. Previous studies of unbiased quantum many-body techniques with Hubbard interactions on GPUs demonstrate a peak acceleration of around an order of magnitude for Determinant Quantum Monte Carlo (DQMC) [22] and Density Matrix Renormalisation Group (DMRG) [23], between one and two orders of magnitude for Hybrid Monte Carlo (HMC) [24] and Exact Diagonalisation (ED) [25], two orders of magnitude for a quantum Monte Carlo impurity solver for the Dynamical Mean Field Theory (DMFT) [26]. This is alongside a rich history of acceleration of classical problems, such as molecular dynamics simulations [27].

By porting the recent combinatorial summation (CoS) algorithm for generic Feynman-diagrammatic expansions [10] to GPUs we achieve at least a three orders of magnitude speed-up. The CoS technique employs a directed graph to evaluate the sum of all integrands of a given expansion order $n$ in a factorised form, so that, e.g., for connected Feynman diagrams, all $\mathcal{O}(n!)$ terms are summed in only $\mathcal{O}(n^2 2^n)$ floating-point operations. It is the layered structure of data processing in the CoS technique, identical to that in a neural network (see Fig. 1), that makes this algorithm amenable to significant hardware acceleration on GPUs.

In contrast to central processing units (CPUs), GPUs are optimised for performing simple linear algebra operations in hundreds of threads in parallel following a common set of instructions. Whereas individual CPU nodes in HPC systems typically possess less than a hundred

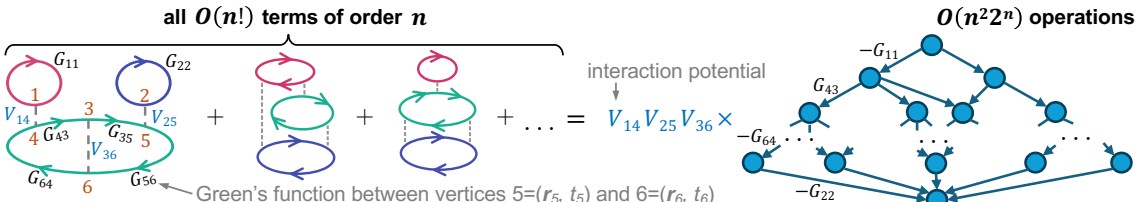

Figure 1: Illustration of the technique for combinatorial summation (CoS) of the integrands of all connected Feynman diagrams of order $n$ by means of a directed graph [10]. The terms are constructed from the Greens functions $G_{\alpha\beta}$ and interaction potentials $V_{\alpha\beta}$ between vertices $\alpha, \beta$ with the coordinates in space-imaginary time $\alpha = (\mathbf{r}_\alpha, t_\alpha), \beta = (\mathbf{r}_\beta, t_\beta)$. The top node of the graph has the value 1; each node accumulates a sum of all contributions from its incoming edges; each edge transfers the value of its origin node to the destination node multiplied by the corresponding Green's function; the bottom node gives the result of the calculation.

general-purpose and sophisticated processor cores, GPUs have many thousands of densely packed smaller cores with shared access to data and instructions. The interconnectivity and sheer quantity of processors that GPUs offer allow for very efficient scaling of parallel algorithms [1], provided that they are based on simple and invariable instructions applied to large volumes of similarly-structured data, as is the case of the CoS technique.

GPUs are also much more energy efficient in performing common floating-point operations than traditional CPU clusters. A good rule of thumb is to expect a GPU to use an order of magnitude less power than an equivalent CPU configuration [29–31]. For floating-point operations with double precision (FP64), top-of-the-line CPUs achieve around 0.024 TFLOP/s per Watt (for AMD EPYC 9654), whereas the efficiency of GPUs can reach as high as 0.15 TFLOP/s per Watt (for NVIDIA H100 NVL on its FP64 "Tensor Core" architecture). If a reduction to quarter-precision (FP8) is permissible, this figure further rises to 8.35 TFLOP/s per Watt, allowing for GPUs' energy efficiency of $\sim 350$ times that of best CPUs. For parallelisable algorithms, one can also expect the cost of purposefully designed hardware to be significantly less than that of assembling large numbers of CPU cores in a cluster for an equivalent data output rate. Indeed, we find that for fast Feynman diagrammatics GPUs can be on the order of a magnitude more cost effective than simply purchasing more CPU nodes.

## 2 Parallelisation of Feynman diagram summation

### 2.1 Mental model of parallelism

A good mental model for parallel programming is to imagine parallelism as a sequence "vectorised" operations. In the simple case, this could just be addition or multiplication, but the same picture holds for the whole suite of typical operations such as memory transactions and evaluating boolean conditions. Consider for example the dot product between two large $n$-dimensional vectors $c = \vec{a} \cdot \vec{b}, \vec{a} = (a_1, \ldots, a_n), \vec{b} = (b_1, \ldots, b_n)$. The serialised implementation of the dot product might perform one index of the summation per iteration $i$ of a loop, $c = \sum_{i=1}^{n} a_i b_i$, requiring $n$ steps in total. In contrast, suppose that we have some parallel architecture that can execute $k < n$ operations simultaneously. It could then take $nk^{-1}$ steps to perform the contraction, $c = \sum_{j=1}^{nk^{-1}} c_j$, with each of $c_j = \left[ \sum_{i=1}^{k} a_{(j-1)k+i} \, b_{(j-1)k+i} \right]$ evaluated

---

[1]We are also witnessing the advent of unified CPU-GPU 'superchips' which continue to erode the current downsides of hardware acceleration (limited memory and throughput between host and device) [28].

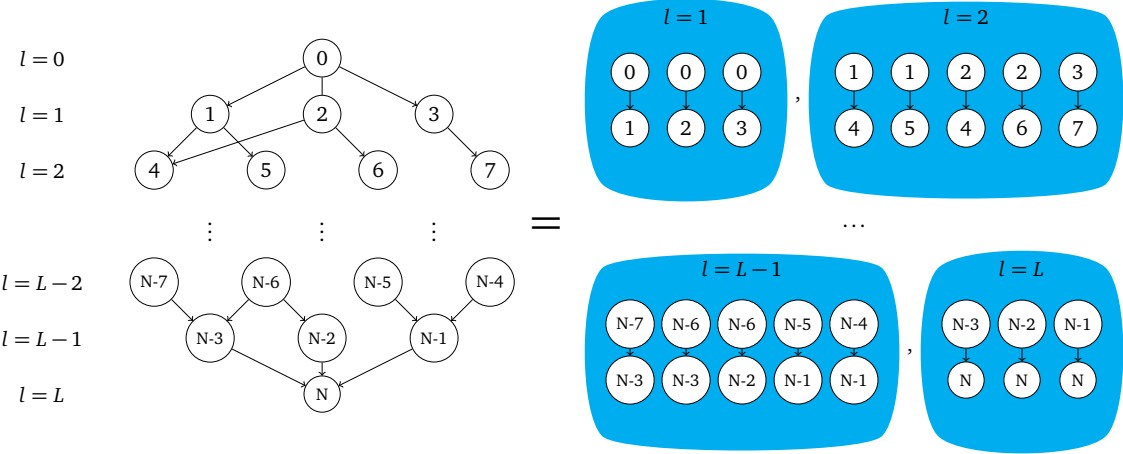

Figure 2: Illustration of the graph flattening transformation for parallel processing.
*LHS:* Example of several levels of a digraph with *L* many levels and *N* many nodes.
The sum of all diagrams is accumulated into the final node *N* as the total of the con-
tributions from each path through the graph. *RHS:* The same graph, now represented
as a flattened array of edges. Each blue box represents a level of the graph, the edges
of which can be processed in parallel.

in one step, so that $c$ takes a total of $2nk^{-1}$ steps to compute. This is advantageous because,
even if each of the $k$ parallel contractions individually take up to a few times longer to exe-
cute than a serial one, at $k \gg 1$ the parallel evaluation is still faster. Specialised hardware
architectures allow this parallel "bandwidth" $k$ to be as large as $\sim 1000$, significantly reducing
the execution time if used correctly. This concept is familiar in the context of modern CPUs
as "Single Instruction, Multiple Data" (SIMD). However, the GPU execution model is known
as "Single Instruction, Multiple Threads" (SIMT), a subtle difference which we expand upon
in Appendix A.1. For our purposes, it suffices to say that we have a set of threads which all
execute a common set of instructions over different data. This set of threads therefore are our
parallel bandwidth $k$, as introduced above.

## 2.2  CoS Algorithm

The CoS algorithm, introduced in Ref. [10], is a powerful machinery for calculating the exact
value of the sum of integrands of all $n$-th order Feynman diagrams for a given set of inter-
nal coordinates. The result is constructed explicitly by factorising it into sums of individual
Greens functions $G_{\alpha\beta}$ between the given set of vertices in space-imaginary time, $\alpha = (\mathbf{r}_\alpha, t_\alpha)$,
$\beta = (\mathbf{r}_\beta, t_\beta)$, times the corresponding interaction potentials $V_{\alpha\beta}$. The factorised sum is thus
evaluated by means of a directed weighted graph, whereby the initial node holds a value of
unity, each edge multiplies the value at its root node by a specific Green's function (propaga-
tor) and adds this result to the node at its head, while the node at the last level of the graph
accumulates the net sum of all diagram contributions; see Fig. 1 for an illustration. A unique
path through this graph, the number of which is combinatorial in the diagram order $n$, pro-
duces one diagram, and yet their sum is evaluated in a dramatically fewer than $\mathcal{O}(n!)$ number
of operations because each edge is shared by a multitude of diagrams. Such a graph for a given
set of Feynman diagrams is not unique and should be optimised to minimise its size, i.e. its
computational cost. Currently, all connected diagrams of order $n$ can be summed in $\mathcal{O}(n^2 2^n)$
operations [10].

The integration over the vertex coordinates can then be performed by stochastic (Monte
Carlo) sampling. Early DiagMC algorithms [3–6] generated individual diagrams one by one,

stochastically sampling not only the space of vertex configurations, but also the space of all
diagram topologies. The deterministic evaluation of the integrand at an exponential cost beats
the factorial scaling of the space of topologies and, for sign-alternating diagrams (as is the
case for fermions), enables cancellations between the terms at the level of the Monte Carlo
configuration weight, significantly reducing the statistical variance, which generically scales
exponentially with $n$ [11].

The CoS approach has further advantages for its practical implementation. Firstly, the
highly-factorised diagram sum provides more numerical stability through avoiding subtrac-
tions of big numbers, allowing for the use of mixed or low precision data types without incur-
ring large losses in the precision of the final result. Secondly, the graph for each expansion
order of a given observable only needs to be constructed once before any calculations are per-
formed. The overall structure of these graphs is also much the same, usually varying only
by size and specific level-to-level linkage between different kinds of expansions and diagram
orders, meaning that the acceleration infrastructure has only to be built once.

## 2.3  Parallelising CoS

**GPU architecture and programming interface.** Here we work in the framework of the CUDA
application programming interface for NVIDIA GPUs, and as such our discussions concerning
implementation are specific to this ecosystem; however, the same concepts may be applied to
other GPU parallelism constructs such as AMD's ROCm and HIP, OpenCL, OpenACC, or CPU-
based parallelism for example with OpenMP. For those who are unfamiliar with the CUDA
programming model, we introduce the required material here and provide a more complete
overview in the Appendix A.

The basic GPU architecture that CUDA exposes to programmers is a large set of threads,
divided up into a user-defined grid of blocks. These blocks may contain up to 1024 threads
each, and share a common pool of local memory, but are comprised of sub-blocks of 32 threads
known as "warps". The warp is the smallest unit of parallelism within CUDA, and the basic
unit of SIMT execution; whilst one is able to request blocks of smaller sizes, each block will still
occupy one warp. With a few exceptions, every thread in a warp executes the same instructions
simultaneously, much like a wave arriving at shore, hence the name for AMDs equivalent to
warps — "wavefronts". How these resources are allocated is a highly problem-dependent task,
e.g. for solving differential equations in a large 2D domain, a natural choice may be to launch
many 2D blocks of 32x32 threads each, whereas other cases may demand smaller 1D blocks
of e.g. 32 threads alone.

**Graph flattening.** The key observation is that each edge on a given level of the graph is
completely independent from its neighbours, an important fact given that the limiting factor
of the algorithm is the exponential growth of the number of edges. The generic operation we
need to perform upon each edge is both simple arithmetic, comprising of one addition and
(up to) 2 multiplications, and is near-identical across the whole graph. Thus, we have a large
number of repetitive arithmetic tasks and can use the panoply of threads at our disposal in
modern GPUs to handle each one in parallel. The number of edges per node at a given level
is not fixed—depending on the diagrammatic rules of the given series, some nodes will have
many links whilst others will have few—meaning that node-based parallelism would be a poor
choice as this would impose warp divergence through uneven workloads across threads. With
this in mind, we treat the edges as the first-class object in the graph, rather than the nodes,
and traverse this large set in parallel. An example of the data structure that meets this need
is given in the right hand side of Fig.2 where we have node-agnostic collections of edges that
may be processed concurrently. The nodes then become nothing more than memory locations
to be read from and written to.

This "flattening" transformation of the graph also provides improved memory access pat-

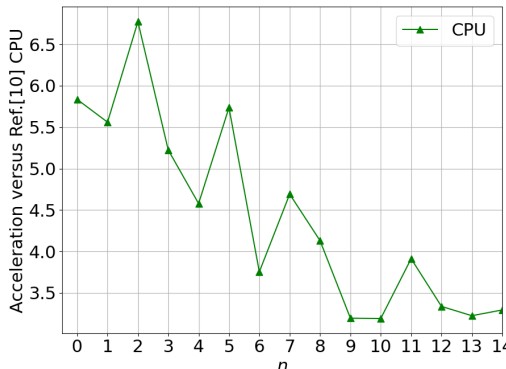

Figure 3: Speed-up seen from transforming the graph to the "flat" representation illustrated in Fig. 2 on the CPU relative to the original implementation of Ref. [10], demonstrating significant acceleration just by a difference in implementation detail, which is achieved here due to a more favourable memory access pattern.

terns, which by itself already leads to an acceleration of the graph evaluation of the order of 5× on the CPU, before any parallelism is employed. Fig. 3 shows the relative speed of a flattened CPU implementation of the CoS algorithm against the code used in Ref. [10].

**Windowing.** There are many different routes from which to approach the task of deciding how to best divide up the power of a large GPU, but this is largely a problem-dependent question. In our case, the task becomes a question of how to walk this flattened data structure in an efficient manner. One natural choice is to scan through the graph in large batches of contiguous edges, thus we assign each graph evaluation to one block of threads and task each thread with implementing one edge operation in Fig. 2 on a given level: multiplication of the value stored in the root node by a specific Green's function and adding the result to the target node.

With the goal to evaluate as many edges on a given level as possible in parallel, we usually choose the number of threads in the user-defined block to be the smallest multiple of 32 that surpasses the average number of edges on a level of the current graph, but does not exceed the maximum allowed block size of 1024. It is usually preferable to choose the smallest block size that can cover a level at once, as this allows more concurrent blocks to be resident on the device to saturate it with work. Where this block cannot cover the entire breadth of a level, as is the case at higher diagram orders, computing all edges can be performed by a successive shifting and re-application of the thread block before the next level may be evaluated. We shall therefore refer to one block of threads as a "window", which must scan through the array of edges, processing all edges within the window at once. This idea is illustrated in Fig.4.

In the simplest implementation demonstrated here, we require that the parent node be completely evaluated prior to each edge operation, which is just a statement that each level of the graph should be evaluated sequentially. This is justified in the limit of large graphs (already reached by $n \gtrsim 7$) where in many levels the number of edges is greater than the thread window size and thus we lose proportionally little computing power by evaluating partially-filled windows at the end of such levels. That is, for the types of graph where we require the most acceleration, this approach minimises the amount of synchronisation required whilst maintaining simplicity of implementation since the GPU is already well-saturated with work. For further accelerating smaller- to intermediate-size graphs, the approach to windowing could be improved by allowing asynchronous evaluation of different graph levels.

**Distributing the task on a grid of blocks.** There are typically tens of thousands of physical

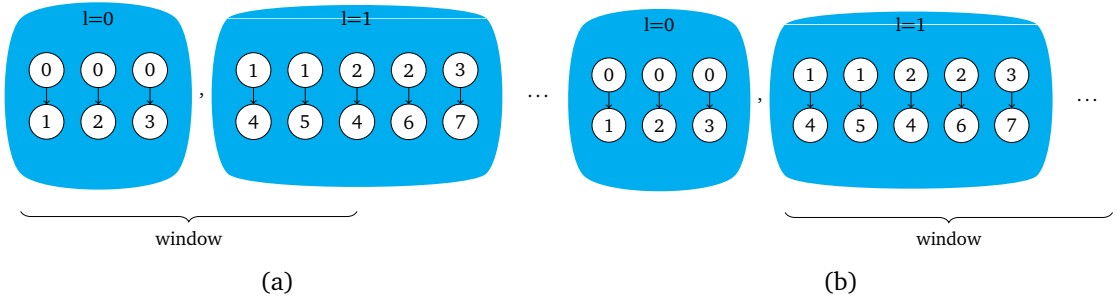

Figure 4: Illustration of how a window iterates through the flattened-representation of the graph. If the size of the level is larger than the window we have to perform a move within the level. (a) Position of the window during the first step through a graph evaluation. (b) Window position after having computed the first level, shifting to the start of the second level.

cores on modern GPUs, such that many thread blocks can be running simultaneously in 1D, 2D, or 3D arrays, with the extent of these grids being limited to at most $(2^{31} - 1)$ blocks. We thus must organise the calculation on the larger scale. Since we have fully leveraged parallelism within a thread block, it is reasonable to use an independent load distribution approach on the scale of the block grid, whereby evaluation of a single graph is managed by a single block and the many blocks in the grid evaluate a large number of graphs simultaneously, each for a different set of internal variables. This tactic is natural for Monte Carlo calculations, where a large number of samples needs to be collected, and is identical to the usual approach of running many Markov chain walkers across different nodes on an HPC cluster, but instead localised to a single GPU node with the benefit of data localisation. The same approach is applicable in methods for deterministic evaluation of the integrals over their internal variables, such as, e.g., the Tensor Train (TT) technique [32], which require many integrand evaluations in order to machine-learn the high-dimensional integrands.

## 3 Results

**Graph evaluations per second.** We benchmark our code with a test of how many function-calls of the graph of a given diagram order $n$ each implementation can complete per second. Fig.5 displays the results for the flattened CoS algorithm proposed here and executed on a CPU (labelled 'CPU') and three different Nvidia GPUs against the performance of the current state-of-the-art CPU code used in prior work ('Ref. [10] CPU'). A core finding is that during the time that the original algorithm sums all diagram integrands for order $n$, the hardware-accelerated flattened CoS evaluates that for an expansion order greater by approximately 8. This corresponds to a doubling of the typically-accessible diagram order, and vastly extends the range of tractable problems.

**Acceleration.** The performance data of Fig. 5 implies a significant speed-up of the series evaluation relative to the original CPU code of Ref. [10]. The corresponding acceleration factor is plotted for each of the implementations in Fig.6. Here we see a significant performance uplift for all cards tested, achieving three orders of magnitude acceleration with the H100 across most orders tested. We also see acceleration factors of $\mathcal{O}(100)$ even when computing low-order expansion terms since we can easily have on the order of thousands of configurations in flight at once, despite not optimising our implementation to best suit the smaller problem size of these tasks.

We might expect to see a flat acceleration curve over all diagram orders based upon the fact

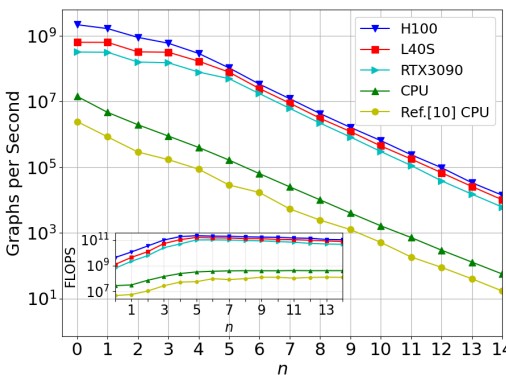

Figure 5: Number of evaluations of the sum of all diagram integrands of order $n$ per second for several types of hardware. The original implementation of Ref. [10] executed on a state-of-the-art CPU is labelled as Ref. [10] CPU. All other data are obtained with the flattened graph of Fig. 2 on the same CPU (labelled CPU) and Nvidia GPU cards: RTX3090, L40S and H100, using single-precision arithmetic. Data was also taken for the A40 GPU, however it is not resolvable from the RTX3090 on this scale. Inset: The corresponding number of floating point operations per second (FLOPS) at each diagram order $n$.

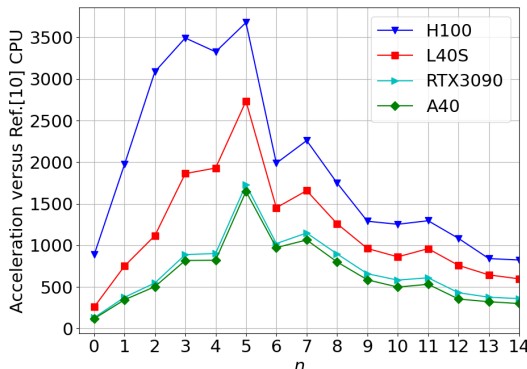

Figure 6: Acceleration factor achieved on the specified GPUs relative to the implementation of Ref. [10] executed on a CPU that corresponds to the data of Fig. 5.

that a GPU is able to perform the same operations as a CPU, just spread over many thousands of threads. From Fig. 6, this is clearly not the case. We see a steep rise to peak performance several times that of the asymptotic scaling of the algorithm for intermediate diagram orders. There appears to be two competing effects that contribute to this. Firstly, we achieve a low-order exponential speed-up versus the serial version of the algorithm. This is explained by low-order graphs having many levels which can be evaluated by a single step of the window (effectively in constant time), and given that there are $\mathcal{O}(n)$ levels in each graph, the GPU takes a linear amount of steps for these smaller graphs whilst the CPU has to compute the exponentially growing number of edges sequentially. Once the graphs grow such that the average size of each level surpasses the maximum window dimension, we see a decay away from the initial region of exponential speed-up since the GPU can no longer process the graph in a linear number of steps. This decay is seen before the apparent saturation in part because each graph beyond $n \sim 5$ requires an exponential amount of window repositioning and memory

transactions, with the corresponding time overheads now proportional to the graph size rather than being only linear in diagram order. Secondly, for very small graphs there is an under-utilisation of each block due to how diminutive the levels of these graphs are. Often here there are more threads inactive than active, in fact it is not until order 4 where the average number of links per level grows past the size of one warp. This is reflected in the inset of Fig. 2, where the number of floating point operations per second (FLOPS) carried out during each of the graph evaluations in the main plot do not saturate until roughly order 5. Therefore, in the regime of $n \lesssim 5$, much of the potential performance is wasted by under-filling of each window. One way to combat this is to design a separate kernel which can process several small levels in a single block to improve concurrency, but this is not a critical issue since the bottleneck of practical calculations is at the highest orders.

**Comparison of devices** Notably, we observe similar performance between the RTX 3090 and it's server-grade alternative A40, so much so in fact that their performance curves are near identical and we opt not to include both for the sake of clarity. This behaviour is reasonable given that they share the same architecture, and have a similar number of Streaming Multi-processors (SMs, see Appendix.A.2), but the newer cards from the Hopper (H100) and Ada Lovelace (L40s) generations pull far ahead in our testing. Both the H100 and L40S have over 50% more SMs than these Ampere cards, and whilst direct comparison of hardware between different generations of architecture is not straightforward, this is evidently reflected in our data by the substantial increase in speed between the Ampere and later cards. Whilst for graphs of moderate order (up to at least $n \sim 10$) the amount of global memory is not directly tied to the speed of execution since we reach saturation in concurrency before running out of memory, having more space at our disposal is always preferred since the storage required for the sums of large graphs can reach many megabytes, placing a cap on how many graphs may be evaluated at once.

The biggest winner in terms of price-to-performance is the RTX 3090, a card that was released for a third of the price of the A40 and yet performs approximately as well as it. The 3090 was designed primarily for the consumer gaming and creative markets, and as such has a limited 24GB of global memory available compared to the 48GB of the A40 and L40S, however, as we have discussed, this does not appear to be a problem for typical cases. The significance of this is that a single desktop workstation, equipped with one or more off-the-shelf gaming cards, could provide a comparatively inexpensive solution for accelerating diagrammatic codes—obtaining the volume of data comparable to that produced in the same time by a typical HPC cluster—along with the fact that there is likely more performance to be gained from the high-end cards in taking better advantage of their substantial memory capacity.

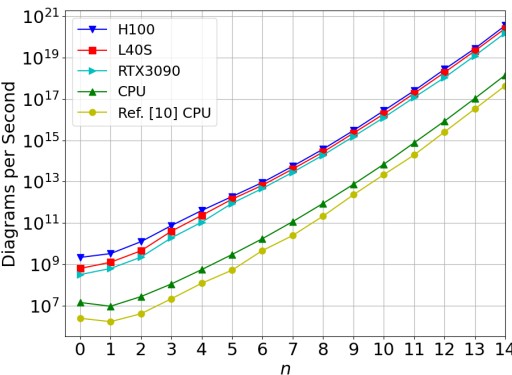

Figure 7: Number of connected Feynman diagrams evaluated per second at each diagram order $n$ by the CPU and GPU implementations in Fig. 5.

**Number of diagrams per second.** The rate at which we are able to do work is hard to appreciate in terms of the evaluation of directed graphs. Instead, a good way to showcase the method in familiar terms is to examine how many Feynman diagram integrands we are evaluating at once per second by the accelerated CoS algorithm. As shown in Fig.7, we are able to sum far beyond a billion billion diagrams each second at high orders and this number is an increasing function of the diagram order $n$. This should be compared with the evaluation of the diagrams individually, one by one, the cost of each and their number growing with $n$ as $\mathcal{O}(n)$ and $\mathcal{O}(n!)$ respectively.

**Numerical stability** The approach also appears to be strongly robust against numerical instability resulting from reduced-precision data types, which is due to the effective factorisation of the large number of terms evaluated in the graph level by level [10]. When repeatedly evaluating the same order 10 configuration we find that there is on the order of $10^{-5}\%$ error accumulated from a single-precision (32-bit floating point) calculation, compared to the standard in scientific computing calculation with double-precision (64-bit) data type. Not only is this strikingly small for such a large computation, but in practice this difference is negligible given the usual error associated with numerical integration of these integrands.

In general we are constrained not by the capacity of the GPU to do calculations, but by the rate at which data can be accessed on the device. In order to evaluate one edge of the graph, we must load the indices of its tail and head nodes, the value from the tail, the relevant Green's Function for the multiplication, and check whether we need to additionally multiply the result by $-1$ if we are closing a fermion loop. Much of this information is shared between nearby edges, which amortises the cost of each fetch, but there is inherently little re-use of data, meaning that the device is often waiting for memory rather than performing useful work. The advantage therefore of moving to smaller 'width' types (e.g. 64-bit to 32-bit floating point) is not just that we are able to fit more computations at once onto the GPU, but a of reduction of the pressure on the memory bandwidth of the device, meaning that we see an additional performance uplift of up to 80% at intermediate orders. This further boost to the speed of our calculations, at the minimal cost of fractional percentage numerical error, is significant enough to justify the use of single-precision calculations throughout.

# 4 Discussion

Alongside the previously discussed dip in speed for low-orders, our peak performance is still below the peak output of the devices tested in this study, we observe that there is up to two orders of magnitude in raw computing power, for e.g. the H100, still on the table when comparing our peak FLOPS versus the theoretical maximum output claimed by Nvidia, clearly the data-intensive nature of the problem still remains challenging even with judicious design. One avenue to pursue is the Tensor Core architecture on Nvidia GPUs, which is capable of significantly higher throughput of matrix operations through dedicated hardware in each SM and the use of reduced and mixed precision data types.

As is demonstrated by the significant speed-up achieved here, diagrammatic methods such as CoS appear to be very well-suited to acceleration with GPUs.

The principal reason for the success of this method is that the operations involved in evaluating the graph for a given configuration are very naturally expressible in GPU-only code, there need not be any reliance upon time-intensive CPU-GPU communication as the entire computation can operate on the device in a self-contained manner. Second, the structural and topological information of the graph may be separated from the value associated to each edge, meaning that the graph needs only be stored once and from this we may evaluate many different configurations simultaneously, saving a significant amount of memory when compared

to matrix-based approaches such as those in [23, 25, 26].

Another factor for the strength of this method is that there is a relative "simplicity" of the tasks that the GPU needs to perform. The whole device is effectively just performing repetitive arithmetic on a large set of data without dependencies of the operations, the precise task that GPUs were designed to perform. At most, if one chooses Monte Carlo sampling as their integration method, then one needs to handle proposal and acceptance of configurations; however, this provides very little overhead in our testing given that Monte Carlo schemes can be readily implemented in device-sided code with the "cuRAND" library from Nvidia.

The highly factorised graph structure provides us with strong numerical stability but also allows for the use of reduced-precision types. This offers an avenue to go beyond single-precision with data-types, such as 'TF32' on recent Nvidia cards, which purportedly offer an order of magnitude faster calculations versus usual FP32 types in machine-learning and linear-algebra contexts [31].

A single node can host several GPUs, one of the most common configurations being eight devices per node, each of which can be running independent calculations or be unified together with technology such as NVLINK. Even in the case of just a single device per node, this allows for one self-contained workstation to take the place of hundreds of traditional CPUs, making precision many-body calculations accessible even when HPC clusters are unavailable.

An interesting observation is that the topology of our graphs is precisely the same as those commonly used in machine learning contexts, such as multi-level perceptrons (MLPs). Indeed, these MLPs are directed graphs which perform the same multiply and accumulate procedure that we use (along side other operations such as activation functions). However, in contrast to many typical MLPs in use, our graphs grow to very large and irregular proportions. This makes optimisations that rely on fixed and or small width levels, such as those in [33], impractical for our uses. A recasting of our graphs into a form that is more amenable to this type of acceleration would provide significant benefits due to eliminating the overhead of window movement.

Due to the growing popularity of new hardware accelerators designed specifically for AI and machine learning use cases, one open question lies in whether existing infrastructure for machine learning could be taken advantage of to provide a platform for fast diagrammatic calculations. Neural processing units (NPUs) are an emerging new class of accelerator that are being deployed in both consumer and cloud-based offerings for the purpose of accelerating inference of tasks such as large language models (LLMs). Since our procedure is surprisingly similar to the task of inference on certain neural networks, native hardware support for the kind of large multiply-and-accumulate operation that we perform is very promising. Field programmable gate arrays (FGPAs) have also seen a rise in use in the realm of machine learning, however the cost of acquiring and the task of programming them is sufficiently high that they may remain useful only to those who have the very precise requirement for them. Lastly, CUDA is but one of several established methods for leveraging parallelism in scientific computing, and whilst it provides a demonstrably powerful set of tools, there exist many other parallelisation constructs - such as the previously mentioned AMD HIP and OpenCL - which allow cross-platform development for GPUs from different vendors, and as such are worth investigation depending upon the hardware available.

So called "parsimonious tensor train" representations of the sum over Feynman diagrams, through the use of tensor cross interpolation (TCI) [32], are a promising new alternative to traditional Monte Carlo sampling of high-dimensional integrals, demonstrating $\mathcal{O}(\frac{1}{N^2})$ convergence of the integration over the internal variables, where $N$ is the number of function evaluations. However, successful applications of this approach remain limited to few-body systems such as the Anderson impurity. Full many-body lattice or continuum-space systems pose a challenge potentially due to the entanglement between multiple space and time coordi-

nates and the computational cost of evaluating the integrands. The significant acceleration of diagram evaluation for generic systems demonstrated here and the construction of the unsymmetrised (over the internal vertices) and thus intrinsically less entangled integrand [10] in the CoS approach addresses these problems. Additionally, linear algebra, and generically tensor algebra, operations, are very fast to perform with parallel execution, therefore we expect that a GPU-accelerated TCI implementation is a very promising route.

# 5   Acknowledgements

J.S. is grateful to Mike Giles and Wes Armour for their excellent course on CUDA programming.

**Funding information**   This work was supported by EPSRC through Grant No. EP/X01245X/1. The calculations were performed using King's Computational Research, Engineering and Technology Environment (CREATE).

# A   CUDA Introduction

Here we give a brief introduction to some of the foundational ideas in GPU parallelism, and specifically the CUDA programming model.

## A.1   Execution Model

In the typical model of computation, a set of instructions (e.g. arithmetic operations) are executed in a single sequential stream known as a thread. Each thread therefore processes one instruction at a time over a single element of a set of data, leading to the common name "Single Instruction, Single Data" (SISD). Most modern CPUs are more sophisticated than this simple construction however, and can execute across $\mathcal{O}(10)$ threads concurrently, but the concept remains the same.

One of the most ubiquitous types of parallelism is "Single Instruction, Multiple Data" (SIMD) wherein one operation (e.g. addition, subtraction, memory transaction) is applied concurrently over a whole set of data, usually by means of dedicated vectorised hardware which each thread can address. The benefit of this model is that highly-decoupled operations such as vector addition can be performed trivially at approximately the cost of a single operation. This type of parallelism is often available on contemporary CPUs, and can be programmed with explicitly using architecture-specific intrinsics.

"Single Instruction, Multiple Threads" (SIMT) in contrast to SIMD allows many threads to perform common sets of instructions over disparate data in parallel, and is the foundation upon which GPU parallelism is built, however there is typically incentive to ensure that there is parity in the instructions and locality of data which these threads act upon. During the execution of code on a GPU, all available threads could operate in lockstep, scheduled to perform the same instructions on their own individual sets of data, or be running several separate tasks, such as different kernels operating in a pipeline model.

In the typical nomenclature, the GPU is often referred to as a "device", whereas the CPU is known as the "host". Subroutines written to be executed by the GPU within this paradigm are known as "kernels", and free-functions which are used by kernels are often known as "device functions".

## A.2 SMs and Occupancy

The CUDA programming model provides several layers of abstraction from the "bare metal" hardware, but it is crucial for good performance to keep in mind the underlying architecture. Device kernels require, at minimum, the number of threads requested for each block, and how many of these independent blocks to launch. Due to the physical limitations of real hardware, there exists a set of interdependent constraints upon the size of each thread block and how many blocks may be simultaneously resident on the GPU. A choice of larger blocks naturally limits the amount of them that may be in flight at once, hence, these two numbers are often free parameters which are tuned to produce the best performance. These constraints are formed by the partitioning of the GPU into Streaming Multiprocessors (SMs), each SM manages the scheduling and execution of its own set of threads, local memory, and registers, and as such can be thought of as an individual parallel processor. Individual thread blocks therefore must draw all of their resources from a single SM, a requirement that places the aforementioned upper bound on the size and number of blocks allowed. SMs are further subdivided into "warps" which are the principal unit through which the parallelism is accomplished. Each warp contains exactly 32 threads, each of which execute synchronously, therefore even if one requests a block size that is not a multiple of 32 - a whole number of warps will still be active. Just as blocks share a common set of local memory, threads within a warp may perform an operation known as a "shuffle" to exchange data directly between themselves inside registers, allowing for very low-latency implementations of tasks such as reduction by avoiding both GMEM and SMEM. Each SM hosts 2048 threads, or 64 warps, whereas each block may be composed of at most 1024 threads. This places an upper limit on the occupancy of the largest blocks as two times the number of SMs available on the device.

## A.3 Synchronicity

Thread execution within each warp occurs at the same time - however with branching and conditional code there may be additional synchronisation involved. Each warp within a block is not guaranteed to execute in order of their 'index', and neither is there any guarantee over execution order between blocks - unless the programmer implements their own synchronisation operations through the use of e.g. atomic locks. There exist several constructs within CUDA to force blocks, or indeed the whole device, to synchronise; however, this is often at the cost of performance since it places restrictions upon how much shuffling around of tasks the scheduler can perform in the background. Parallel and/or concurrent code is often at its fastest when it can be expressed in a wait-free, or almost wait-free, manner.

## A.4 Memory Hierarchies

There exists two main addressable regions of memory on CUDA devices. Firstly, global memory (often referred to as GMEM). This is the main bulk of memory available on the GPU, when cards are advertised as having a capacity of 48GB, for example, it is the GMEM which is being referred to. GMEM is addressable from all threads on the device, and is allocated by invocation of host-sided subroutines. Both the large capacity and the global visibility of GMEM comes at the cost of it being the slowest memory resource available on the device, often requiring on the order of hundreds of clock cycles to return requests from, and therefore it is best practice to minimise the amount of transactions each kernel performs with GMEM.

Shared memory (SMEM), on the other hand, is localised within each SM and is restricted to capacities on the order of 10s to 100s of kilobytes per SM. This localisation enforces that each region of SMEM is unique to each block, meaning that blocks have no visibility of other blocks' SMEM, but comes with the benefit that it is an order of magnitude faster than GMEM.

This means that SMEM is excellent for caching data which a block will need frequent access to, preventing expensive calls out into GMEM. For our use-case, this space in SMEM is ideal for storing the matrix of Green's Functions, from which the graph's edges are drawn, along with the current and previous vertex configurations. There also exists a small amount of "constant" memory in each SM, which allows programmers to keep commonly used small parameters, such as simulation bounds or uncommon numerical constants, as close to the threads as possible - again minimising the amount of long-range memory transactions that need to occur.

Not only do memory transactions to/from distant locations impose wait times an order of magnitude longer than that taken to perform arithmetic operations, but this scaling also applies to the energy cost for these operations, therefore we wish to minimise the amount of times that we exchange or fetch data. Counter-intuitively, it is often more efficient to re-calculate quantities, rather than store and load them when needed.

## A.5  Cache and Locality

The message of the previous sections is that data locality and re-use are crucial to the performance of GPU algorithms. In order to write fast and energy-efficient code we therefore need to be mindful of our memory access patterns and how this translates to cache utilisation.

It is best practice to layout memory in a manner that satisfies spatial and temporal locality. In other words, keep associated data close together such that it is likely to be fetched together in one cache line. This is especially important in the context of GPU programming because warps will fetch memory for all of their threads at once, if the layout of this memory is poorly optimised then it is likely that this could lead to cache misses or incur many times as many fetches than actually required to fit the absolute size of the requested memory. One way we can target this is by avoiding object-orientated paradigms, such as fine-grained encapsulation, by preferring "Struct of Arrays" over "Arrays of Structs".

Another important concept is coalesced memory accesses. The scheduler coalesces each warps memory requests into the fewest possible number of operations, therefore, if we design our access patterns with this in mind we can help minimise this number. Suppose that we have a block of 32 threads which loops over an array of $32^2$ doubles in GMEM. If we were to write a traditional for-loop, where thread 0 processes elements $\{0, 1, \ldots, 31\}$, thread 1 processes elements $\{32, 33, \ldots, 63\}$, and so on until the end of the array, each thread would require 8 bytes from locations separated in GMEM by 256 bytes - much larger than the typical cache line size of 128 bytes, therefore necessitating 32 reads per loop iteration. A much improved, but counter-intuitive, method is for each thread to index into the GMEM array by its index plus multiples of 32. Therefore, thread 0 processes elements $\{0, 32, \ldots, 992\}$, thread 1 processes elements $\{1, 33, \ldots, 993\}$, etc, such that elements $\{0, 1, \ldots, 31\}$ are all available on the first loop iteration, $\{32, 33, \ldots, 63\}$ on the second, and so on until the end of the array is reached. Each thread still receives the correct 8 bytes, however the warp can fetch all 256 bytes in just 2 reads per loop iteration since the (128 byte) cache lines will be full of useful, coalesced, data, vastly improving performance.

## A.6  Host-Device Latency

Just as with memory transactions, we want to minimise long-distance or high-latency communications, and this includes the Host-Device channel. Most systems interface the host and device with PCIe connections, which while suitable for most graphics use cases can be a significant bottleneck for scientific applications which require a large amount of data flow between the CPU and GPU, being often an order of magnitude slower again than addressing GMEM. This problem is mostly unavoidable, but is mitigated by the use of smart algorithm design and modern interfaces. Nvidia offer alternative interconnection solutions such as; SXM for

greater bandwidth between host and device, NVLINK for direct device-to-device interfaces, and even unified "super-chips" which offer a GPU and CPU on a single board with common high-bandwidth memory pools.

Ultimately however, the task still falls upon the programmer to circumvent much of this issue by writing self-sufficient device code to avoid this latency. CUDA provides the useful abstraction of "streams" which allow for execution and host-device communication to be performed in parallel, amortising much of the cost of these transactions.

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
