# Peer review of "Exploiting Parallelism for Fast Feynman Diagrammatics"

_SciPost Physics_

## Round 1 · Referee Report · Anonymous (Referee 1) · 2025-3-17

Strengths

1-a very impressive acceleration in the evaluation of Feynman diagrams
2-this is turn makes the computation of higher orders more affordable and can thus help in getting more accurate results or reach lower temperature.
3-the speedups are well illustrated and documented, and the text is well written.
4-a comparison between different graphical cards (and CPU) is given so that readers can estimate what gives them currently the most cost-effective options

Weaknesses

1- it is a technical report, there is no physics in it and the work builds entirely on the previously published CoS algorithm by one of the authors. To what extent some of the conclusions will be relevant with the next generation of GPUs, is hard to say but I have my doubts. 2- unclear what readers of the paper take from it: without possessing an implementation of the CoS algorithm it is hard to appreciate the technical and GPU specific increases. Also for non-CUDA experts this paper might be hard to digest. 3-although the speedup is dramatic the performance is still 2 orders of magnitude below peak performance. This is good and not good at the same time, and reasons are given for that, but it leaves the reader with the question if there is room for further improvement.

Report

(see strengths and weaknesses)

Requested changes

1-in what sense do the technical advances enable new physics? Adding one example where the speedup leads to a significant change in the evaluation of an observable would make a strong case 2- it is unclear to me what is in the git repo. All I could find is the data for the figures and corresponding python plot but that does not seem very relevant to me? 3-I disagree with the statement on line 33 that the only systematic error is the maximum order n. When series are asymptotic or divergent and we have no analytical understanding (as is often the case), then the maximum order is certainly not the only sytematic error 4- line 100: can the degree of speedup through parallellization be quantified theoretically and compared to the one obtained?

Recommendation

Ask for minor revision

---

## Round 1 · Referee Report · Anonymous (Referee 2) · 2025-4-7

Strengths

1- Strong GPU acceleration for Feynman diagram evaluations 2- clear demonstration of scaling properties in figures and cleanly written text

Weaknesses

1- no detailed code provided 2- while a large number of samples is great, a discussion regarding the number of samples required to generate good estimates of measurements. n=14 is great but can you get a target error estimate? 3- the dependence on the COS algorithm is logical, but there are not many research groups pursuing this direction 4- lack of physics demonstration.

Report

I believe that it is possible, that if the authours follow-up this work with a number of physics-based papers, that a publication of this sort will garner quite a few citations.

Requested changes

1- a physically justified example problem to address the samples vs error estimate problem. I don't expect you can get reliable measurements of n=14 diagrams this way. Though to be fair, I don't think there is an alternate way that does so. 2- cleaner definitions of variables when used in figures - consider redefining them in captions 3- more details on what is actually implemented. An algorithm block, with pseudocode that helps orient what specifically your code is doing would be... perhaps essential. Otherwise it is not reproduceable.

Recommendation

Publish (meets expectations and criteria for this Journal)

---

## Editorial Decision

awaiting_resubmission